# Antenatal Determinants of Childhood Obesity in High-Risk Offspring: Protocol for the DiGest Follow-Up Study

**DOI:** 10.3390/nu13041156

**Published:** 2021-03-31

**Authors:** Danielle Jones, Emanuella De Lucia Rolfe, Kirsten L. Rennie, Linda M. Oude Griep, Laura C. Kusinski, Deborah J. Hughes, Soren Brage, Ken K. Ong, Kathryn Beardsall, Claire L. Meek

**Affiliations:** 1MRC Epidemiology Unit, University of Cambridge, Cambridge CB2 0QQ, UK; Danielle.Jones@mrc-epid.cam.ac.uk (D.J.); soren.brage@mrc-epid.cam.ac.uk (S.B.); Ken.Ong@mrc-epid.cam.ac.uk (K.K.O.); 2Institute of Metabolic Science, University of Cambridge, Cambridge CB2 0QQ, UK; lck34@medschl.cam.ac.uk (L.C.K.); deborah.hughes8@nhs.net (D.J.H.); 3NIHR Cambridge Biomedical Research Centre—Diet, Anthropometry and Physical Activity Group, MRC Epidemiology Unit, University of Cambridge, Cambridge CB2 0QQ, UK; Emanuella.De-Lucia-Rolfe@mrc-epid.cam.ac.uk (E.D.L.R.); Kirsten.Rennie@mrc-epid.cam.ac.uk (K.L.R.); Linda.OudeGriep@mrc-epid.cam.ac.uk (L.M.O.G.); 4Department of Paediatric Medicine, University of Cambridge, Cambridge CB2 0QQ, UK; kb274@cam.ac.uk; 5Cambridge Universities NHS Foundation Trust, Cambridge Biomedical Campus, Cambridge CB2 0QQ, UK

**Keywords:** gestational diabetes mellitus, pregnancy, study protocol, randomised controlled trial, large for gestational age, complex intervention, calorie restriction, maternal weight gain, childhood obesity, adiposity, type 2 diabetes, prevention

## Abstract

Childhood obesity is an area of intense concern internationally and is influenced by events during antenatal and postnatal life. Although pregnancy complications, such as gestational diabetes and large-for-gestational-age birthweight have been associated with increased obesity risk in offspring, very few successful interventions in pregnancy have been identified. We describe a study protocol to identify if a reduced calorie diet in pregnancy can reduce adiposity in children to 3 years of age. The dietary intervention in gestational diabetes (DiGest) study is a randomised, controlled trial of a reduced calorie diet provided by a whole-diet replacement in pregnant women with gestational diabetes. Women receive a weekly dietbox intervention from enrolment until delivery and are blinded to calorie allocation. This follow-up study will assess associations between a reduced calorie diet in pregnancy with offspring adiposity and maternal weight and glycaemia. Anthropometry will be performed in infants and mothers at 3 months, 1, 2 and 3 years post-birth. Glycaemia will be assessed using bloodspot C-peptide in infants and continuous glucose monitoring with HbA1c in mothers. Data regarding maternal glycaemia in pregnancy, maternal nutrition, infant birthweight, offspring feeding behaviour and milk composition will also be collected. The DiGest follow-up study is expected to take 5 years, with recruitment finishing in 2026.

## 1. Introduction

Gestational diabetes, a common complication of pregnancy, is associated with short-term and long-term health implications for the baby [1,2]. Infants are commonly large-for-gestational-age at birth (LGA; >90th centile) and have a higher risk of obesity in childhood [2,3]. Unfortunately, very few interventions are available with proven efficacy to reduce the likelihood of childhood obesity in these high-risk children. The early development of obesity in children with existing environmental and genetic susceptibilities to type 2 diabetes is a major public health concern [4].

Events in pregnancy, perinatal and early postnatal periods may be important for future childhood obesity, but are relatively understudied, particularly in specific high-risk populations [5]. Babies born to mothers with gestational diabetes often have multiple risk factors for childhood obesity, which appear to have an additive effect upon risk. Maternal obesity in pregnancy [6,7], maternal excessive gestational weight gain [6], maternal postnatal weight retention [6], exposure to hyperglycaemia in utero [8,9,10], perinatal complications such as large-for-gestational age at birth [9,11,12], infant formula feeding [13] and increased growth trajectory in early life [14] are all established risk factors for childhood obesity or adiposity and are common features of a pregnancy affected by gestational diabetes.

It is therefore possible that an intervention which addresses maternal weight in pregnancy may reduce obesity rates in offspring. Family interventions (which target at least one parent to improve obesity rates in children) are already well-established in the prevention of childhood obesity [5]. However, many interventions to reduce the risk of childhood obesity target older children (2–10 years) and may miss the opportunity to intervene in early years [5,15]. The DiGest study, a currently ongoing dietary intervention in pregnant diagnosed with gestational diabetes, provides the opportunity to study the influence of a pregnancy weight intervention upon risk factors for childhood obesity in early life [16].

The DiGest Study is a randomised, double-blind controlled trial of a reduced calorie diet using a novel dietary intervention to assess the benefits of controlling maternal weight gain in late pregnancy in gestational diabetes. The trial is described in detail elsewhere [16]. Briefly, women are randomised to receive a weekly dietbox containing all meals and snacks and are blinded to the overall calorie content (1200 kcal/day for intervention group and 2000 kcal/day for control group; 40% carbohydrate, 25% protein, 35% fat). The dietbox commences at enrolment, typically 28–32 weeks’ gestation, and continues until delivery of the infant. The clinical care team and research team are also blinded to calorie allocation. Dietboxes are nutritionally balanced and low in glycaemic index, low in saturated fat, high in vegetables and protein and suitable for use in pregnancy. Data will be collected to assess the impact upon maternal weight gain, infant birthweight and a range of obstetric and glycaemic outcomes during late pregnancy up to 3 months postnatally [16]. The design of the DiGest trial provides the opportunity for a controlled and blinded dietary study and reduces potential bias due to differences in maternal educational level, cooking ability, income and kitchen facilities.

In this manuscript we describe a follow up study to the DiGest trial which investigates the effect of the reduced calorie dietary intervention in pregnant women upon the development of obesity in a high-risk population of children from birth to 3 years of age. The hypothesis is that a reduced calorie diet in late pregnancy in women with gestational diabetes reduces offspring adiposity and improves maternal weight at 1, 2 and 3 years postpartum.

## 2. Materials and Methods

Study design and ethical approval: The DiGest Follow up study is an observational study on the effects of a multicentre, prospective, randomised double-blind controlled dietary intervention trial conducted in late pregnancy. In summary, participants of the follow up study will have been exposed to either the intervention diet of 1200 kcal/day, or the control diet of 2000 kcal/day as part of the DiGest study. Macronutrient ratios were identical for each diet; 40% carbohydrate, 25% protein, 35% fat. Meals were prepared from the same recipes, with a factor of 1.667 used to convert portion size to obtain meals of two different sizes. This diet it provided from enrolment (typically 28–32 weeks’ gestation) to delivery of the infant. Participants will have attended 4 study visits in total to provide blood samples, blood pressure, body weight and anthropometry measurements, and to complete a series of questionnaires. Randomisation for the original DiGest study was stratified for centre. Throughout the DiGest intervention and Follow-up study, both mothers and children will receive standard NHS care, as described in the NICE guidelines [17]. The study is being conducted in accordance with the Declaration of Helsinki, and the protocol has been submitted to the Research Ethics Committee (UK Bloomsbury REC 21/PR/0213) and the NHS Health Research Authority (IRAS 281062).

Recruitment: The DiGest trial recruitment occurs at 5 hospital trusts in East Anglia, UK. The same study sites will be used for the follow-up study. At 3 months postpartum, DiGest trial participants will be given information about the follow-up study and invited to participate by the research midwives, nurses, clinical research staff or by their physician/obstetrician. For training purposes, students in healthcare disciplines (e.g., medicine, biomedical science, nursing, midwifery) may also occasionally recruit patients under appropriate supervision. Informed consent will be obtained at the final visit of the DiGest dietary intervention, with the mothers providing consent on their infant’s behalf. Participants (or mother-infant dyad) can withdraw from the study at any time without reason without affecting their clinical care. There is no financial incentive for this study, but a small token of appreciation is provided for the child at each visit in line with guidelines of the Royal College of Paediatrics and Child Health [18].

Eligibility criteria: All women from the DiGest cohort (confirmed gestational diabetes and BMI 25 kg/m^2^ or above at enrolment) are eligible to enrol in the follow-up study, however, they must be recruited within 12 months of the baby’s birth. Mothers would be excluded if they are unwilling or unable to provide informed consent, if they experienced stillbirth, neonatal death or had an infant born with severe congenital anomaly.

Follow-up visit structure: The study timeline is outlined in Figure 1. Study visits will be carried out in the participants’ home, local hospital or at another place convenient for the participant. The initial follow-up visit will coincide with the final DiGest visit at 3 months after the birth, where the consent form will be signed for both the mother and infant. Maternal and infant anthropometry will be measured at this visit as part of the DiGest study. Further Follow-up visits will take place at 1, 2 and 3 years postnatally and will take approximately 45 min.

Anthropometry Measurements: At all visits, maternal height and weight will be measured using a routinely calibrated stadiometer and weight scale (Seca Hammer Steindamm, Birmingham, U.K.). Waist and hip circumference will be measured to the nearest 0.1 cm with a fibreglass tape, in accordance with the World Health Organisation criteria [19]. Waist circumference is located at the midpoint between the lowest palpable rib and the iliac crest. Hip circumference is measured at the greater trochanters, or at the widest extension of the buttocks. Other maternal anthropometry that will be measured include mid upper arm circumference and skinfold thickness, using Harpenden calipers recorded to the nearest 0.2 mm. Infant length and weight will be taken in a supine position, measured to the nearest 0.1 cm using fibreglass tape, and 0.01 kg using scales (SECA 757 Infant digital scale, Seca, Birmingham, UK). Infant abdominal circumference, head circumference, skinfold thickness (Holtain calipers, Crosswell, Wales, U.K.), mid upper arm circumference will also be measured by trained research staff according to methods described elsewhere [16]. All equipment used to measure anthropometry are routinely calibrated.

Maternal Glucose Assessment: Due to the COVID-19 pandemic, a home-based OGTT using continuous glucose monitoring (CGM) will replace the gold standard OGTT for assessment of maternal glucose tolerance postnatally. An HbA1c will also be performed to replace mothers’ annual diabetes check in primary care. We have previously assessed the feasibility and efficacy of the home-based OGTT with good results (Kusinski et al., submitted to press). In brief, a Dexcom G6 CGM sensor is sited during the study visit with a masked receiver so participants do not see their glucose results in real time. On day 3, participants are asked to eat normally, and fast overnight for at least 10 h. On the morning of day 4, at 09.00, participants are asked to drink a sachet of Rapilose (Galen, Craigavon, UK) containing 75 g of anhydrous glucose. Participants can have sips of water but are asked to consume no other foods or drinks for 3 h after the test. The timing of the home OGTT is chosen to coincide with peak sensor accuracy. Glucose readings are taken automatically every 5 min and transmit to the CGM receiver. Results from the OGTT at 0, 1 and 2 h are included in the analysis. Other CGM metrics will also be used to assess glycaemia as described in a recent CGM consensus statement. CGM metrics will be reported using both adult non-pregnant and pregnant ranges to allow comparison with pregnancy data gathered in the DiGest trial (also using a Dexcom G6 system).

### Physical Activity Assessment

Participants will be asked to wear a wrist-worn accelerometer continuously for 7 days concurrently with the CGM. The triaxial accelerometer is waterproof and does not have a visual display, nor any auditory or vibrational cues, which means that participants will not be able to influence their activity level based on what is recorded by the device and nor will they be prompted to move about during periods of inactivity. These accelerometers have been used in in women during and after pregnancy to assess their daily physical activity with high compliance and produce reliable estimates of energy expenditure, overall physical activity and moderate-vigorous intensity activity [20,21,22]. Accelerometry data at 100 Hz will be collected and downloaded from the monitors for analysis. At the end of the recording period, mothers will be asked to complete the Recent Physical Activity Questionnaire (RPAQ), a self-completion questionnaire designed to assess an individual’s physical activity over the previous four weeks. The questionnaire contains questions about physical activity in four domains: at home, at work, commuting and during leisure time. RPAQ has been validated against doubly labelled water and individually calibrated heart rate and movement sensing to assess physical activity energy expenditure (PAEE) in adults [23,24]. It has been used in diabetes prevention trials [25] and in longitudinal studies of pregnant women [26].

Other Biochemistry samples: A blood spot sample will be taken from mothers and frozen at −80 °C for future batch analysis of C-peptide and metabolomics. An optional heelprick blood spot will also be taken from infants, for future batch analysis of C-peptide and metabolomics. If a genetic sample has not been taken already as part of the DiGest trial, a cheek swab will be taken from both mothers and infants. Mothers will be asked to provide a sample of milk (formula or breast) which will be collected onto filter paper for assessment of infant nutrition including lipidomic profiling. To protect participant’s privacy, this can be performed after the visit.

Questionnaires: Mothers will be asked to complete validated questionnaires about quality of life (EuroQuol EQ5D), eating behaviour (three factor eating questionnaire—TFEQ-18) [27], physical activity (RPAQ) [24] and web-based multiple pass 24 h dietary recalls to assess habitual dietary intake (Intake24; [28,29]). These questionnaires have been used during the DiGest trial and participants will be familiar with them. In addition, mothers will be asked to complete questionnaires about parental feeding style (PFSQ) and their baby or child’s eating behaviour (CEBQ) [30,31,32,33]. Information will be collected about infant feeding choice and if relevant, duration of breastfeeding.

## 3. Results

The aim of the study is to investigate the effects of a reduced calorie diet in late pregnancy in women diagnosed with gestational diabetes upon longer-term maternal and offspring metabolic outcomes. The primary outcome for child health is standardised weight at 1, 2 and 3 years of age. The primary outcome for the maternal population is maternal weight at 1, 2 and 3 years postpartum.

Offspring secondary outcomes at 1, 2 and 3 years of age: There are multiple secondary outcomes for children including weight, BMI, growth trajectory, and blood spot biomarkers such as C-peptide or metabolomics at 1, 2 and 3 years. Questionnaire data will be assessed to identify effects of the intervention in pregnancy upon child eating behaviour, with assessment for confounding factors including maternal BMI, maternal eating behaviour and parental feeding style.

Maternal secondary outcomes at 1, 2 and 3 years postpartum: Maternal outcomes to be studied include maternal weight and weight change, BMI, anthropometry measures of adiposity, glycaemia (CGM metrics, HbA1c, OGTT results, indices of insulin production and sensitivity, including HOMA-IR and HOMA-B, Matsuda score and Stumvoll index [34,35], cardiometabolic health (blood pressure, heart rate, lipids, fasting insulin, fasting glucose), maternal food intake, food nutritional content and quality, eating behaviour, quality of life, and incidence of type 2 diabetes or gestational diabetes in a future pregnancy.

Analysis Plan: An intention to treat analysis of the primary outcome for child health (standardised weight at 1, 2 and 3 years of age) will be based on linear regression with adjustment for the stratification variable of study centre through a fixed effects model. The potential role of other explanatory variables such as pre-pregnancy BMI, infant nutrition, infant postnatal growth trajectory or information from the questionnaires will be investigated. A per protocol analysis will also be performed in participants with >80% compliance and at least 4 weeks’ exposure to the intervention. Secondary outcomes will also be examined through regression analyses (linear or logistic) appropriate for the type of outcome being considered.

Power calculation: All eligible women and their infants will be invited to join the follow-up study. However, calculations are based on assuming a 50% recruitment rate (*n* = 250 women and their infants) and a 20% withdrawal rate. For the maternal primary endpoint, data from earlier work suggest that typical values for maternal BMI outside of pregnancy in women with a history of gestational diabetes is mean 28.7 kg/m^2^ (SD 7.1; *n* = 416) and maternal postpartum HbA1c 37.5 mmol/mol (SD 7.5; *n* = 157) [36]. Using these figures, recruitment of 250 women, will give 90% power to identify a 3 kg/m^2^ difference in BMI (e.g., 29 vs. 32 kg/m^2^) and a 3 mmol/mol difference in HbA1c postnatally while allowing for a 10–20% withdrawal rate. At 80% power, this sample size is sufficient to identify a 2 kg/m^2^ difference in BMI (e.g., 30 vs. 32 kg/m^2^) and a 2 mmol/mol difference in HbA1c postnatally.

For the offspring primary endpoint, assessment of infant weight will be based upon z-(SD) scores. At the sample size of 250 infants, there will be 90% power to identify a 0.45 SD increase in weight with 80% power to identify 0.4 SD increase in weight. At the age of 2 years old, a z-score of 0.4 is equivalent to 0.5 kg.

## 4. Discussion

This follow-up study of the DiGest randomised controlled trial provides a unique opportunity to assess the potential benefits of a dietary intervention in late pregnancy upon the development of obesity in children with multiple risk factors. The availability of data from mid pregnancy until the age of 3 years also allows detailed characterisation of the relative importance of pregnancy and postnatal risk factors in the development of adiposity in early childhood.

Rates of maternal obesity are increasing in the antenatal population throughout the world, and pre-pregnancy BMI is a strong predictor of both birthweight and future childhood obesity. A recent metanalysis identified that maternal obesity was significantly associated with overweight/obesity in early, mid and late childhood with odds ratios 2.43, 3.12 and 4.47, respectively [6]. Weight gain in pregnancy is also important and has repercussions for women’s BMI for 15 years or more after the pregnancy [37]. Landon and colleagues found that gestational weight gain was strongly related to obesity in children aged 5–10 years old [10].

In addition to the effects of maternal obesity, exposure to intrauterine hyperglycaemia appears to further increase the risk of childhood obesity. There is evidence that maternal glycaemia in gestational diabetes is associated with childhood obesity at 10–14 years [2] and altered anthropometry at 5–10 years, favouring obesity [10]. Maternal hyperglycaemia can also indirectly increase childhood obesity rates, by increasing the risk of LGA in offspring. Data from the UK and Canada suggest that childhood obesity rates in LGA infants are at least twice that of children born appropriate for gestational age [11,38]. The exact mechanisms behind these intrauterine exposures and later life obesity are unclear. It is possible that altered placental secretary function, offspring hyperinsulinism and genetic susceptibilities all play a role.

The design of the DiGest and DiGest follow-up studies also allows longitudinal assessment of the effects of other pregnancy exposures upon longer-term offspring growth and health. For example, metformin use in pregnancy has been associated with lower birth weight but increased postnatal catch-up growth, but the consequences of this upon longer-term offspring cardiometabolic outcomes remain less clear [39,40]. The collection of anthropometric measures in offspring exposed to metformin in utero with paired blood samples, and a comparable unexposed control group, provides opportunity to explore this issue in greater depth.

Serum and cord blood stored for biomarkers such as leptin, adiponectin and placental hormones provides opportunities to identify infants at an earlier stage who are at risk of obesity in childhood. Previous work has demonstrated that cord blood leptin levels are associated with pregnancy diet, physical activity and neonatal body composition in a comparable population [41,42]. Cord blood adiponectin has also been associated with body composition effects which may be distinct in male and female neonates [43] and may additionally provide information about neonatal beta cell function [44]. Placental growth factors and metabolic function have also shown relevance for pregnancy outcomes [45,46]. Taken together, it is feasible that biomarkers in cord blood or maternal serum may facilitate early identification of offspring at risk of obesity and diabetes in later life, who could be prioritized for health interventions.

Although maternal physical activity levels in pregnancy and postpartum are likely to be vital for determining offspring habitual exercise levels, relatively few modifiable factors have been identified in children’s physical activity levels in the very young [47,48]. Findings to date suggest that parents’ physical activity levels are associated with children’s activity levels in pre-school aged children and role-modelling by mothers appears to be one of the strongest associations [48]. However, relatively few studies have examined exercise after gestational diabetes in mothers and children. The DiGest Follow-Up study uses both questionnaires and accelerometers to assess physical activity, information which could inform future interventional studies.

Infant feeding and growth trajectory in the first year of life are also important. Although randomised studies of feeding modality in early life are not possible, observational analyses in unselected populations suggest consistent benefits of breastfeeding upon rates of childhood obesity [49,50]. There is also evidence that breastfeeding reduces childhood obesity risk in offspring of mothers with gestational diabetes and obesity [13,51]. Stettler and colleagues reported that similar benefits may persist until adulthood in a study of offspring to age 20 years [52]. The study also includes questionnaires about child eating behaviour, child food preferences and parental feeding style to examine behavioural associations with obesity and feeding behaviour in children aged up to 3 years.

The aetiology of childhood obesity is therefore complex and multifactorial. In infants of mothers with gestational diabetes, multiple risk factors are often evident at birth. Successful interventions are urgently needed to reduce the risk of obesity and future metabolic disease in these high-risk children.

## 5. Conclusions

The DiGest follow-up study provides the opportunity to assess pregnancy and postnatal risk factors for the development of childhood obesity, and to describe the potential impact of a dietary intervention in pregnancy. Early intervention in offspring with existing environmental and genetic susceptibilities to type 2 diabetes will be vital to break the intergenerational cycle of obesity.

## 6. Patents

No patents are relevant to the study described in this manuscript.

## Figures and Tables

**Figure 1 nutrients-13-01156-f001:**
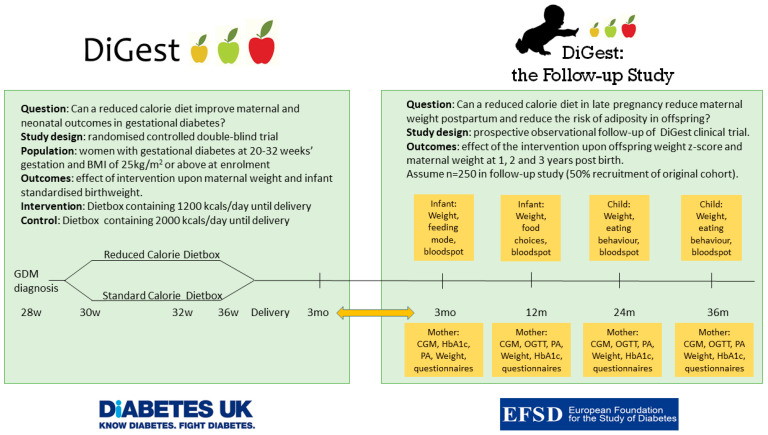
Summary of protocol and links between the DiGest trial and the follow up study.

## Data Availability

No applicable.

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
