# Peer review of "Antenatal Determinants of Childhood Obesity in High-Risk Offspring: Protocol for the DiGest Follow-Up Study"

_nutrients, 2021, doi:10.3390/nu13041156_

Round 1

Reviewer 1 Report

This is a very interesting study protocol to identify whether a reduced calorie diet in pregnant women with overweight/obesity and gestational diabetes can reduce adiposity in children to 3 years of age. The protocol is thorough and well detailed.

Women in this study will be offered treatment with metformin if blood glucose targets are not met with diet and exercise recommendations within 1 to 2 weeks after diagnosis of gestational diabetes, in line with the NICE guidance. Nevertheless, there is evidence suggesting that intrauterine exposure to metformin is associated with neonates significantly smaller than neonates whose mothers were treated with insulin during pregnancy. In addition, despite lower average birth weight, metformin-exposed children appear to experience accelerated postnatal growth, resulting in heavier infants (18-24 months) and higher BMI by mid-childhood (5-9 years) compared to children whose mothers were treated with insulin. Such patterns of low birth weight and postnatal catch-up growth have been reported to be associated with adverse long-term cardio-metabolic outcomes (Tarry-Adkins JL, Aiken CE, Ozanne SE. Neonatal, infant, and childhood growth following metformin versus insulin treatment for gestational diabetes: A systematic review and meta-analysis. PLoS Med. 2019;16(8):e1002848.  doi:10.1371/journal.pmed.1002848).

It would be interesting, if possible, to assess the effect of the reduced calorie diet in patients treated only with insulin versus patients treated with metformin alone or in combination with insulin.

Since metformin exposure appears to affect prenatal and postnatal growth, the results from this study might not be transferable to other countries where insulin is the only pharmacological treatment considered for gestational diabetes treatment.

Author Response

Thank you for the opportunity to submit a revised draft of the manuscript titled
‘Dietary Intervention in gestational diabetes; protocol for the DiGest follow-up study of maternal and child obesity’. We appreciate the time and effort that is required for manuscript review and we have endeavoured to address your points thoroughly. We hope these changes have addressed your concerns and that you now consider the manuscript suitable for publication.

Reviewer 1

Comments and Suggestions for Authors

This is a very interesting study protocol to identify whether a reduced calorie diet in pregnant women with overweight/obesity and gestational diabetes can reduce adiposity in children to 3 years of age. The protocol is thorough and well detailed.

Response: Thank you for your supportive comments.

Comment: Women in this study will be offered treatment with metformin if blood glucose targets are not met with diet and exercise recommendations within 1 to 2 weeks after diagnosis of gestational diabetes, in line with the NICE guidance. Nevertheless, there is evidence suggesting that intrauterine exposure to metformin is associated with neonates significantly smaller than neonates whose mothers were treated with insulin during pregnancy. In addition, despite lower average birth weight, metformin-exposed children appear to experience accelerated postnatal growth, resulting in heavier infants (18-24 months) and higher BMI by mid-childhood (5-9 years) compared to children whose mothers were treated with insulin. Such patterns of low birth weight and postnatal catch-up growth have been reported to be associated with adverse long-term cardio-metabolic outcomes (Tarry-Adkins JL, Aiken CE, Ozanne SE. Neonatal, infant, and childhood growth following metformin versus insulin treatment for gestational diabetes: A systematic review and meta-analysis. PLoS Med. 2019;16(8):e1002848.  doi:10.1371/journal.pmed.1002848).

Response: thank you for this comment. We are aware of this excellent work, which was undertaken by our colleagues in Cambridge. We consider this a very important area for further investigation and have added some related information to the manuscript.

Comment: It would be interesting, if possible, to assess the effect of the reduced calorie diet in patients treated only with insulin versus patients treated with metformin alone or in combination with insulin. Since metformin exposure appears to affect prenatal and postnatal growth, the results from this study might not be transferable to other countries where insulin is the only pharmacological treatment considered for gestational diabetes treatment.

Response: Thank you for this suggestion to explore the effects of the intervention with consideration of insulin or metformin use during pregnancy. We agree that this would make for an interesting analysis and potentially enhance the impact of the study.  We had not mentioned this aspect specifically in this protocol paper, as pregnancy and early postnatal growth will be assessed in our earlier study, DiGest. As part of DiGest, we have planned secondary analyses to assess different growth patterns in women who were treated with diet/lifestyle, metformin only, insulin only or metformin/insulin together.

However, as you have suggested, there is an opportunity as part of the current follow-up study to assess postnatal growth from 3-36 months with respect to metformin use in pregnancy. We have therefore added a section to our results to elaborate on this further.  

Thank you again for your detailed and constructive review which have given us the opportunity to improve out manuscript.

Reviewer 2 Report

The authors took up a very interesting topic. However, in the planned research, I miss the markers of the hormones responsible for lacrimation and tightly related to the fatty tissue. What is more, leptin and adiponectin. In addition, the discussion is little expanded. in my opinion, it requires modification and targeting not only on calories and diet, but also on data from other questionnaires, such as focusing on physical effort.

Author Response

Thank you for the opportunity to submit a revised draft of the manuscript titled
‘Dietary Intervention in gestational diabetes; protocol for the DiGest follow-up study of maternal and child obesity’. We appreciate the time and effort that is required for manuscript review and we have endeavoured to address your points thoroughly. We hope these changes have addressed your concerns and that you now consider the manuscript suitable for publication.

Reviewer 2:

Comments and Suggestions for Authors

Comment: The authors took up a very interesting topic. However, in the planned research, I miss the markers of the hormones responsible for lacrimation and tightly related to the fatty tissue. What is more, leptin and adiponectin.

Response: We are pleased you found our manuscript of interest. We agree that assessment of leptin, adiponectin and lactogenic hormones would be of interest here. These are however moderately expensive analyses, and we don’t have current funding to measure these analytes on samples from all 250 mothers and 250 babies. Blood samples taken during the study will be collected and stored at -80C, which provides the opportunity to carry out a subset analysis once appropriate funding is in place. We have added to the discussion to explain that stored samples may be very useful for future mechanistic studies, or to study biomarkers for future childhood obesity. Thank you for this constructive comment.

Comment: In addition, the discussion is little expanded. In my opinion, it requires modification and targeting not only on calories and diet, but also on data from other questionnaires, such as focusing on physical effort.

Response: Thank you for your comments. We have made the existing parts of the discussion more concise and added brief new information about physical activity and questionnaire data. Following discussion with the team at Nutrients, we discussed our plans with a collaborator, Soren Brage, who is an expert in the field of physical activity (see acknowledgements). We have therefore been able to expand and develop the physical activity assessment planned as part of the study. We anticipate the information gathered will provide an understanding of physical activity patterns and an estimate of energy expenditure, which will be invaluable for our analysis.

Thank you again for your detailed and constructive review which have given us the opportunity to improve out manuscript.